# Fire Safety of External Thermal Insulation Systems (ETICS) in the Aspect of Sustainable Use of Natural Resources

Małgorzata Niziurska *, Michał Wieczorek and Klaudiusz Borkowicz

Lukasiewicz Research Network—Institute of Ceramics and Building Materials, Cementowa 8 Str., 31-983 Krakow, Poland; michal.wieczorek@icimb.lukasiewicz.gov.pl (M.W.); klaudiusz.borkowicz@icimb.lukasiewicz.gov.pl (K.B.)

* Correspondence: malgorzata.niziurska@icimb.lukasiewicz.gov.pl

**Abstract:** External Thermal Insulation Composite Systems (ETICS) are the most common technical solution to improve the thermal efficiency of existing buildings. In the light of the applicable regulations, ETICS are kits that apply only one type of thermal insulation material. All typically used ETICS introduced into the market classify as non-combustible. Despite that, the emerging recommendations in some countries point to the need for the introduction of barriers of non-combustible material such as mineral wool (MW) around the openings and horizontal isolation barriers around the building between different floors to prevent fast fire spread. That raises an important question: Do MW partitions significantly improve the fire safety of the building and balance other aspects such as the sustainability and durability of such combined insulation materials with different properties? Here, we assessed the impact of MW partitions in EPS-based ETICS on the spread of the fire according to the requirements of BS 8414-1: 2020. Four different variants were investigated. The study showed slight differences in average temperatures and the size of the polystyrene melting area for various insulation variants with the introduced horizontal MW partitions. The introduction of MW partitions shows no significant benefits or improvement of fire safety of the investigated ETICS.

**Keywords:** external thermal insulation system (ETICS); expanded polystyrene (EPS); mineral wool (MW); fire performance; fire safety; large-scale fire test



## 1. Introduction

Construction products placed on the European Single Market, the European Union's (EU's) greatest achievement, must meet the basic requirements specified in the Construction Products Regulation (CPR) [1]. One of the basic requirements described in the regulation is to ensure the fire safety of facilities. The other basic requirements of the CPR are: (no. 1) mechanical resistance and stability; (no. 3) hygiene, health and the environment; (no. 4) safety and accessibility in use; (no. 5) protection against noise; (no. 6) energy economy and heat retention; and (no. 7) sustainable use of natural resources. The first five abovementioned requirements and the one mentioned earlier, (no. 2) safety in case of fire, existed in the Council Directive 89/106/EEC (CPD), previously in force in 1989–2013 [2]. In connection with the growing environmental pollution related to the increase in electricity consumption, the production of a vast amount of waste, the shrinkage of raw material resources, and the increasing ecological awareness, a new basic requirement was introduced in CPR: sustainable use of natural resources. The idea behind introducing this requirement is to encourage all building materials producers to develop energy-saving and low-emission technologies limiting the use of fossil resources. The seventh basic requirement regarding the sustainable use of natural resources is defined in various dimensions, namely:

1. The reuse or recyclability of construction works, their materials, and parts after demolition;
2. The durability of construction works;

3.　The use of environmentally compatible raw and secondary materials in construction works [1].

All construction products should be considered through the prism of understanding the sustainable use of natural resources mentioned above. In addition, it is also essential to comprehensively and holistically consider construction products in terms of all basic requirements and find the right balance between them.

Currently used insulation materials for facade insulation are mainly expanded polystyrene (EPS) and mineral wool (MW). They differ in terms of their fire behavior, physical properties, structure, durability, and manufacturing costs. Moreover, one has to consider their life cycle aspects when designing buildings and selecting insulation technology. The environmental impact of thermal insulation materials is well defined [3]. An environmental load of thermal insulation materials has the most significant effect on the environmental impact values of External Thermal Insulation Systems (ETICS)—the most commonly used solution in EU countries that allows the requirements of energy-efficient construction to be met [4]. The influence of the type of render used as the top layer of the ETICS is also well understood [5]. Nowadays, it is crucial to know the aspects related to the safety of use of various materials to be able to consciously choose the optimal solutions, taking into account a wide range of factors, including the issue of environmental impact, which is not subject to mandatory assessment before placing a construction product on the market.

This article explores the results of tests carried out to assess the impact of MW partitions on the spread of fire through a facade with ETICS insulation with EPS.

In terms of fire safety, CPR [1] specifies that construction works must be designed and constructed in such a way that in the event of a fire:

- The load-bearing capacity of the construction can be assumed for a specific period of time;
- The generation and spread of fire and smoke within the construction works are limited;
- The spread of fire to neighboring construction works is limited;
- Occupants can leave the construction works or be rescued by other means;
- The safety of rescue teams is taken into consideration.

Exterior wall cladding can become a contributor to a fire in all these respects.

In the case of a facade system, there are three common scenarios for the occurrence and spread of a fire that one has to consider. The first is a fire inside the building that occurs in a room adjacent to the external wall. Two other scenarios include the case of an external fire spreading due to radiation from an adjoining building or when the fire source is directly next to the facade (balcony, garbage can, car, etc.) [6].

External wall claddings and their components are assessed for their reaction to fire as stand-alone products and systems, according to EN 13501-1 [7], only in the indoor fire scenario. The requirements for fire resistance following EN 13501-2 [8] may include wall cladding as an element of the structure, while the method does not assess the spread of fire on the external surface. Therefore, European Member States use their national test methods. The European Commission has entered into a contract with a consortium led by RISE Research Institutes of Sweden to develop a harmonized European approach to the fire assessment of facades with different requirements, based not only on the spread of fire but also on falling fragments, burning waste, smoldering, smoke, and taking into account details such as windows [9]. Two methods have been proposed, one simply retaining the two existing tests (BS 8414-1 [10] and DIN 4102-20 [11]) and an alternative method. Work is currently underway to develop a familiar concept for assessing the fire resistance of facades.

Large-scale tests are seen as the most representative way of showing the resistance of a facade cladding to fire [6,12,13]. In addition, some numerical modeling methods are available to assess and validate differences between various experimental methods designed for the fire performance testing of existing cladding systems, e.g., BS 8414-1: 2020 [10], SP Fire 105 [14], and ISO 13785-2 [15]. Comparing the experimental measurement with simulation results always meets some uncertainties arising from natural variations in parameters

and the effect of ambient conditions [16–18]. In the case of the BS 8414-1 method [10], those differences were characterized as heat exposure (weight of wood crib), wind, and climate conditions since the tests are performed outdoors. With the use of numerical methods, good agreement between experimental data and the numerical model is observed when measured in the experiment. Heat release rate (HRR) is used as an input in the simulations, excluding the area close to the burning chamber, where higher temperatures in the numerical model were observed [18]. Furthermore, the same parameters are important for the repeatability and reproducibility of experimental results in large-scale testing in accordance with BS 8414-1:2020 [10].

The behavior of the sample during large-scale tests may always differ from the actual scenario and course of the fire. Still, it allows the tests to be carried out in conditions as close to real conditions as possible and through a strictly defined research methodology to compare their results. Therefore, the obtained results can also serve as a basis for comparisons and analyses in the use of belts made of non-flammable materials, which was the idea behind the implementation of this study. Results are crucial in terms of further recommendations and regulations on the design and construction of facade systems for fire safety.

Depending on the height of the building, inter-story partitions, protection over window and door openings, or entire strips of wool along communication routes are used in various EU countries. These solutions have disadvantages that can be seen in the everyday use of the system.

The use of such solutions may result in:

- Cracks in the reinforcement and plaster layers due to different thermal expansion of materials under the influence of external factors, e.g., temperature;
- Discoloration of the facade related to the different absorption capacities of insulation materials.

It is worth noting that the European Technical Assessments, being the basis for introducing ETICS for use, allow the assessment of only one type of insulation material within a given system. In contrast, the combination of various thermal insulation materials within one thermal insulation system has never been the subject of an assessment of the properties or durability of the system by Technical Assessment Bodies [19]. It is not possible to test all product combinations or system variations. Still, it is essential to know that other fire performance or performance characteristics can significantly influence the test result. The safety potential of a facade is highly dependent on which risk is to be assessed. One has to be aware that laboratory tests are performed under defined and controlled conditions under which a standardized fire load is applied. The assessment of the facade's fire safety was the subject of research [6,12,13,19–24], also concerning the use of non-flammable mineral wool partitions.

## 2. Materials and Methods

### 2.1. Goal and Scope

The research aimed to assess the impact of the introduction and arrangement of horizontal mineral wool (MW) partitions in the thermal insulation system made of expanded polystyrene (EPS) on the spread of fire on a large scale under the requirements of BS 8414-1: 2020 [10].

### 2.2. ETICS System Components

The investigated ETICS system (variant I) and its components (variant II–IV) were placed on the market according to the European Technical Assessment ETA 15/0582 issued by the Institute of Ceramics and Building Materials based on ETAG 004 [25].

The investigated systems consisted of two different thermal insulation materials:

- Mineral wool (MW), with properties described by product designation codes T5-DS(80,90)-S(10)20-TR10-PL(5)200-WS-WL(P)-MU1 according to EN 13162: 2012 + A1: 2015 standard [26]);

- Expanded polystyrene (EPS), with properties described by product designation codes T1-L2-W2-Sb5-P5-BS115-CS(10)70-DS(N)2-DS(70,-)2-TR100 according to EN 13163: 2012 + A1: 2015 standard [27].

The components of the system are presented in Table 1. Trade names and detailed characteristics of the components of investigated systems are presented in Supplementary Materials S1–S4. The system was partially bonded with additional mechanical fixing. The strip of adhesive applied along the perimeter was at least 3 cm wide. In addition, 6 to 8 dabs of adhesives, ca. $8 \div 12$ cm in diameter, were distributed evenly on the remaining surface. The bonded surface was at least 40% (60% after application and pressing).

**Table 1.** EITCS system components according to European Technical Assessment ETA 15/0582 [25] used for BS 8414-1: 2020-fire performance test [10].

| Category | Component Description | Quantity per m$^2$ |
|---|---|---|
| Adhesives for EPS | General purpose (GP) cement-based adhesive for EPS bonding, modified with redispersible polymers, and mineral fillers, characterized by compressive strength class CS IV and water absorption class W$_2$ according to EN 998-1: 2016 standard [28] | ca. 7.50 kg |
| Base coat adhesive | General purpose (GP) cement-based base coat adhesive for embedding the reinforcing glass fiber mesh, modified with redispersible polymers, and mineral fillers, characterized by compressive strength class CS IV and water absorption class W$_2$ according to EN 998-1: 2016 standard [28] | ca. 5.95 kg |
| Glass fiber mesh | Alkaline-resistant glass mesh with a nominal weight of 150 g/m$^2$ | ca. 0.15 kg |
| Key coat | Mixed both acrylic copolymers and silicone dispersions key coat with mineral filers and additives use as adhesion primer for silicone finishing coat | ca. 0.45 kg |
| Finishing coat | Thin-layer silicone render with 1.5 mm mineral filler compliant with EN 15824: 2017 standard [29] | ca. 2.90 kg |
| Ancillary materials | Aluminum starter tracks with fixing, corner beads with mesh, universal facade anchors compliant with EAD 330196-01-0604 [30] | - |

### 2.3. BS 8414-1 Fire Performance Test Method

Large-scale fire spread tests were carried out under the requirements of BS 8414-1: 2020 [10] entitled "Fire performance of external cladding system. Test method for non-loadbearing external cladding systems fixed to and supported by a masonry substrate" [6]. The research was carried out in the Łukasiewicz Research Network—Institute of Ceramics and Building Materials in Cracow from 7 September to 28 October 2020. The research standard BS 8414-1: 2020 [10] defines the method of assessing the behavior of non-bearing external insulation cladding attached to the brick wall of the building when exposed to fire under controlled conditions [10]. Such a test represents an external fire source and a fully developed internal fire that spreads, e.g., through window openings leading to exposure of the thermal insulation system to outer flames.

In the test, the fire is simulated by burning wood in a combustion chamber, which produces about 4500 MJ of energy during the 30 min. The fire source affects a fully developed fire with a 3.0 (±0.5) MW fire load. The thermal insulation system was attached to a test wall with a corner structure 9700 (±200) mm high, made of Silka E24S (Xella) silicate blocks with a strength class up to 25 MPa (Figure 1a). Temperature sensors compliant with the requirements of BS 8414-1: 2020 [10] were placed at heights of 2.5 m (L1 temperature measurement line), 5 m (L2 temperature measurement line), and 7.5 m (L3 temperature measurement line) above the upper the edges of the combustion chamber. The diagram of the arrangement of measuring thermocouples on the test wall is presented in Figure 1b.

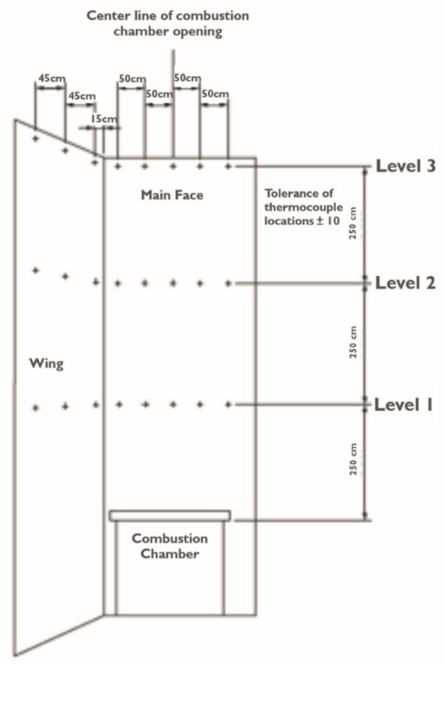

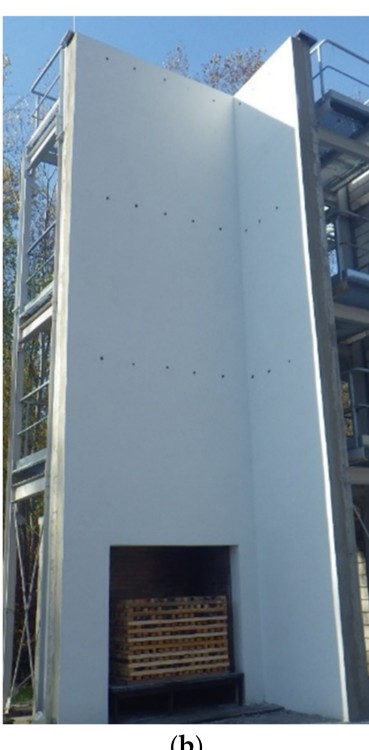

(**a**)                                                                        (**b**)

**Figure 1.** Test wall according to BS 8414-1: 2020 [10] (**a**) diagram showing the arrangement of measurement elements [10]; (**b**) removal of the wall before the test with a visible combustion chamber filled with a properly prepared pile of wood.

The temperature was recorded during the test at three measurement levels (L1–L3), both outside and inside the system. For the L2 and L3 measurement lines, the temperature was measured inside the facade cladding and in the middle of each essential insulation layer. Such arrangement of measuring elements allows for the assessment of fire spread inside the system. According to the assessment criteria and guidelines for the classification of thermal insulation systems BR135 [31], a negative assessment of the tested system takes place if the temperature of one of the external or internal sensors located at level 2 (L2) exceeds the temperature of 600 °C within 15 min from the starting of the test and persists for 30 s. The test start time ($t_s$) is calculated from the moment the initial temperature ($T_s$) is reached at 1 (L1) equal to 200 °C.

The duration of the tests for all samples was 60 min, which consisted of 30 min of exposure to fire and 30 min of observation of the system behavior after extinguishing the fire source. The basis for early termination of the test is (1) the propagation of the flame over the test apparatus, (2) a threat to the safety of personnel, and (3) damage to the measuring equipment.

All system failures, such as complete collapse, chipping, delamination, flaming debris, etc., are recorded during the test. The assessment of the insulation system condition is then included in the overall evaluation of the test sample.

The conducted tests of the degree of fire spreading included various material solutions for the distribution of horizontal partitions in the thermal insulation system and are described further in Section 2.4. Each insulation system was tested only once due to the scale of the test, sample size, and the total cost of the test.

*2.4. Fire Performance Test Variants*

Four variants of an insulated wall were tested according to the standard BS 8414-1: 2020 [10] and according to the following cases:

I.           Thermal insulation made of 150 mm thick EPS, without partitions made of mineral wool—an ETICS system (Figure 2a);

II.          Insulation made of 150 mm thick EPS with a 20 cm wide MW partition located 40 cm above the upper edge of the combustion chamber (Figure 2b);

III.         Insulation made of EPS 150 mm thick with a 20 cm wide MW partition located directly in the lintel of the combustion chamber (Figure 3a);

IV.         Insulation made of 150 mm thick EPS with a 20 cm wide MW partition placed directly in the combustion chamber lintel and a 20 cm wide inter-story partition (Figure 3b).

Variant I of the study concerned an ETICS. Following European legislation, only one thermal insulation material is allowed to be used in ETICS. Variant II of the study concerned a thermal insulation system with an MW separation strip (horizontal partition) classified as A1 in reaction to fire according to PN-EN 13501-1 [7]. A strip of MW was applied across the entire width of the test wall 40 cm above the upper edge of the combustion chamber (20 cm wide strip) (Figure 2b).

In variant III, a horizontal MW partition with A1 reaction to fire class was made in the lintel of the combustion chamber. The strip of MW was 20 cm wide and extended 30 cm beyond the edge of the lintel (Figure 3a). In variant IV, a strip of MW was used in the lintel of the combustion chamber in the same way as in variant III. Additionally, in this variant, an inter-story strip made of MW was used along the entire width of the test wall at a height of 4.8 m above the upper border of the combustion chamber (a strip 20 cm wide) (Figure 3b).

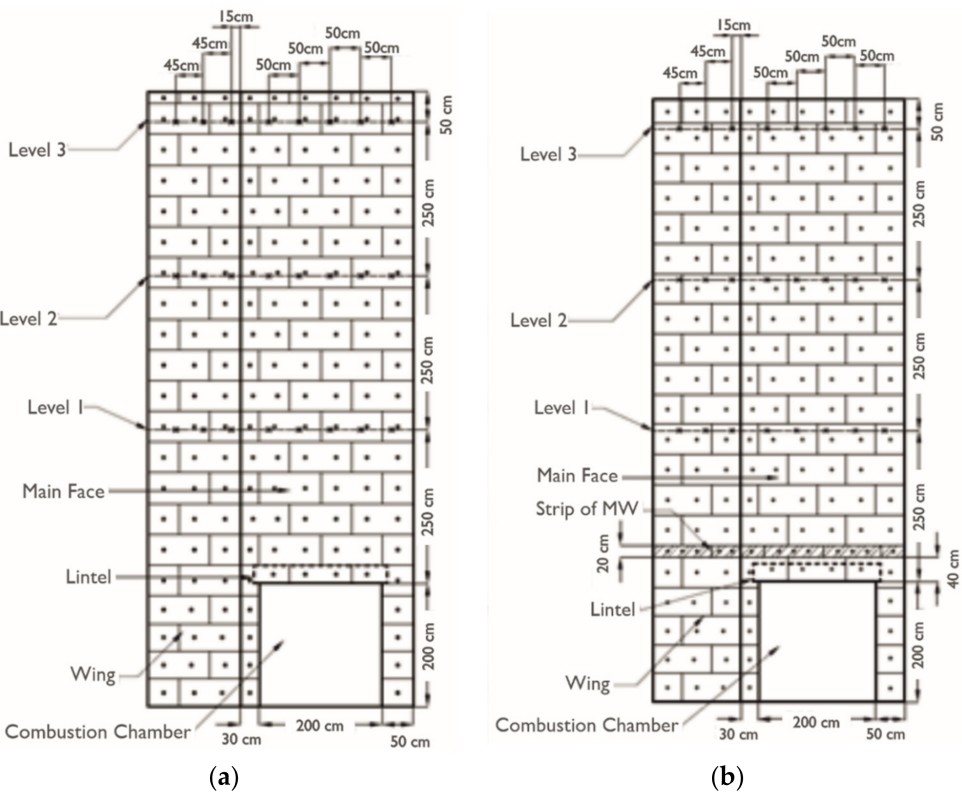

**Figure 2.** Scheme of the test wall: (**a**) ETICS made entirely of EPS—variant I, and (**b**) insulation made of EPS with a 20 cm strip of MW placed 40 cm above the combustion chamber lintel—variant II.

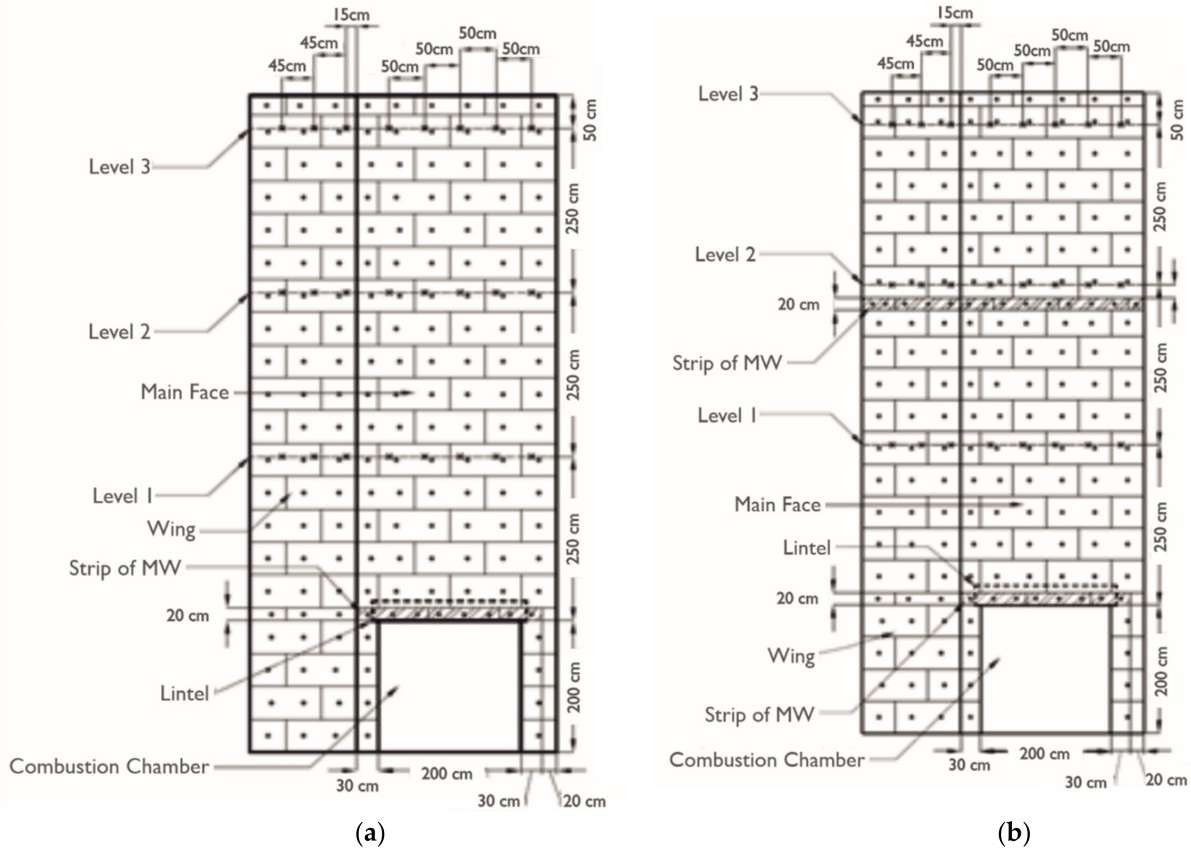

**Figure 3.** Scheme of the test wall: (**a**) insulation made of EPS with a 20 cm strip of MW placed in the overhead of the combustion chamber—variant III; (**b**) insulation made of EPS with a 20 cm strip of MW placed in the lintel of the combustion chamber and a 20 cm inter-story strip of MW located 5 m above the combustion chamber—variant IV.

## 3. Results and Discussion

In the first minutes of the fire test, external thermocouples measured an increase in temperature at level 1 (temperature measurement line L1) in all considered cases. At level 2 (temperature measurement line L2), the temperature increase was observed simultaneously for all tested cases, i.e., when the pile of wood was burning in its entire volume. During this time, no temperature build-up in the reinforced layer and the insulation material occurred.

In the case of variant I, the leakage of molten polystyrene occurred 5 min after the test started due to the crack in the lintel. Molten polystyrene burned further out on the surface of the substrate in front of the combustion chamber (Figure 4a). In the case of variant II of the test wall insulated with EPS with a strip of MW placed 40 cm above the combustion chamber, a crack formed 4 min after the test started (Figure 4b). Flame combustion occurred in the lintel area. Further, single falling flaming drops were also observed. In the case of the remaining variants, there was no presence of falling drops.

Designed solutions represented by variants III and IV showed accumulation of the molten EPS on the strip of MW used in the headroom. Molted EPS promoted further flame combustion, as shown in Figure 4c,d. In the case of variant IV, the reinforced layer detached from the insulation 21 min after the test started. It was due to the accumulation of hot exhaust gases and the increase in pressure under the belt of mineral wool located 5 m above the combustion chamber. The hot flue gases passing above the MW belt caused local detachment of the reinforcing layer from the heat-insulating material, observed as a temporary bulging of the top layer.

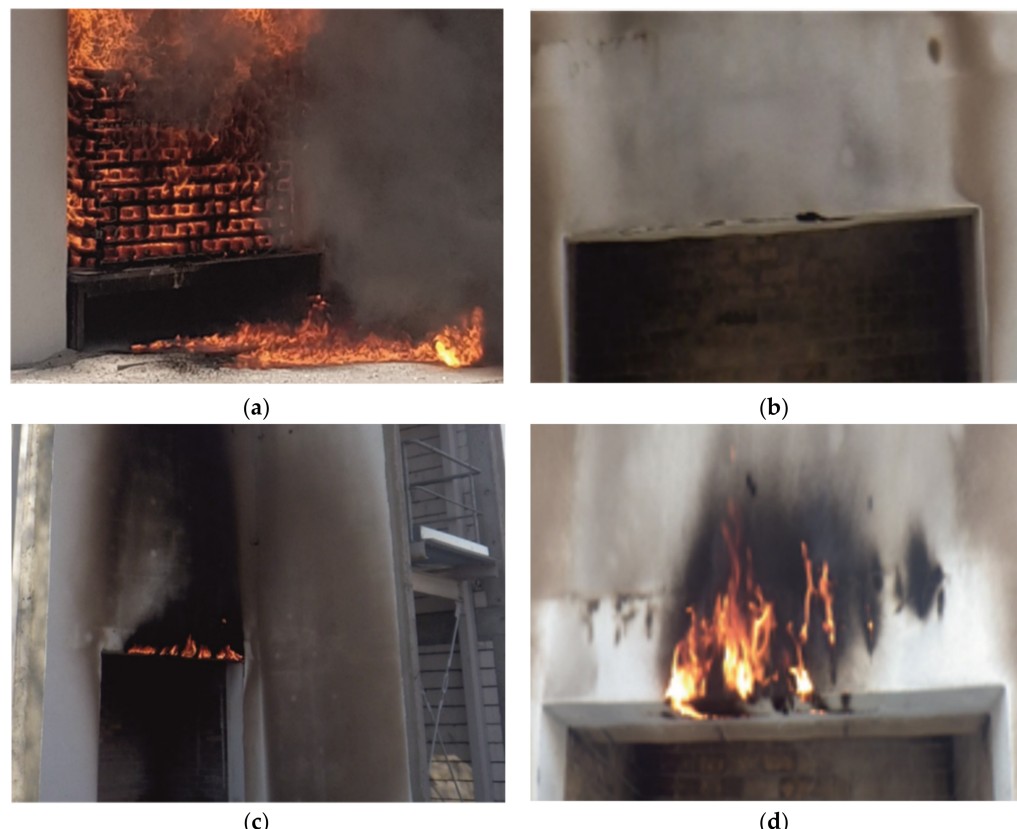

**Figure 4.** (**a**) View of the melted polystyrene in front of the combustion chamber—variant I; (**b**) view of the cracked EPS lintel—variant II; (**c**) view of the burning mineral wool lintel after completion of the test—variant III; (**d**) view of the burning mineral wool lintel after completion of the test—variant IV.

The next part of the work presents the tested walls (variants I–IV) after the end of the test and after removing the reinforced layer (Figure 5a–d).

In all investigated variants, the reinforced insulation layer did not fall off during the test or after the test ended. Despite the EPS melting under the top layer, it was necessary to remove the top layer mechanically at the end of the test. After removing the top layer, the complete melting of the polystyrene on the test wall for variants I to III (Figure 5a–c) was observed. The complete melting of the polystyrene for variant IV (Figure 5d) occurred below the inter-story partition due to the heat deposition. EPS melting below the same MW partition occurred due to the hot gases' formation from the thermal decomposition of exhaust polystyrene. When comparing the photos of variant II (Figure 5b) and IV (Figure 5d), a slight difference in the degree of polystyrene melting above 5 m from the combustion chamber has to be noted. Nevertheless, in both cases, no combustion in this area occurred. It means that no fire spread was observed through the insulation layer regardless of the use of MW.

During the test, the temperature was measured using thermocouples placed in the measurement lines L1–L3. The results are presented as the average temperature values measured by individual thermocouples placed in the insulation material in the L2 measurement line (Figure 6) and the L3 measurement line (Figure 7).

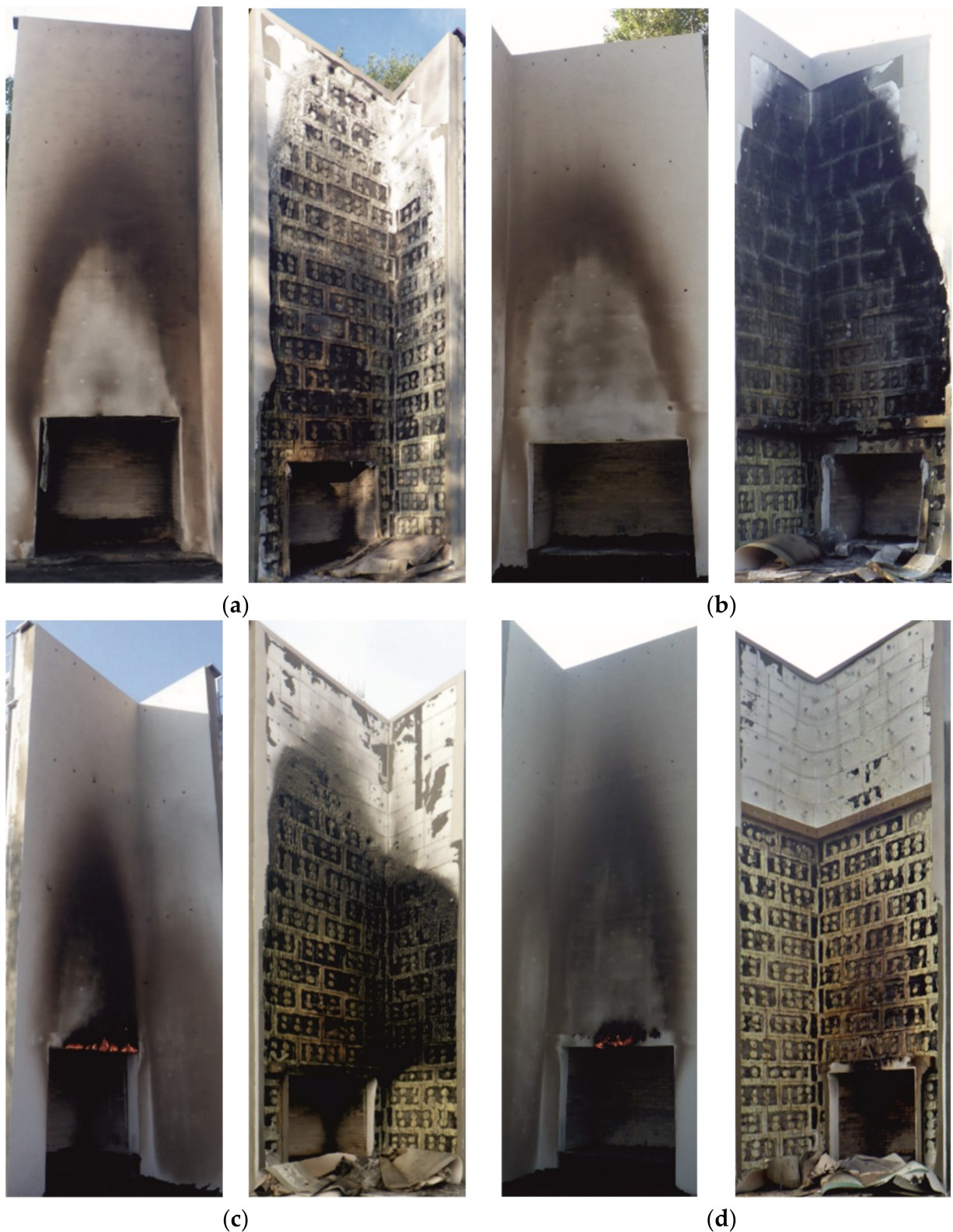

**Figure 5.** Scheme of the test wall after completion of the test and removal of the reinforced layer: (**a**) ETICS—variant I; (**b**) insulation made of EPS with a 20 cm MW belt placed 40 cm above the combustion chamber—variant II; (**c**) insulation made of EPS with a 20 cm strip of MW placed in the lintel of the combustion chamber—variant III; (**d**) insulation made of EPS with a 20 cm strip of MW in the lintel of the combustion chamber and an inter-story strip located at the height of 4.8 m—variant IV.

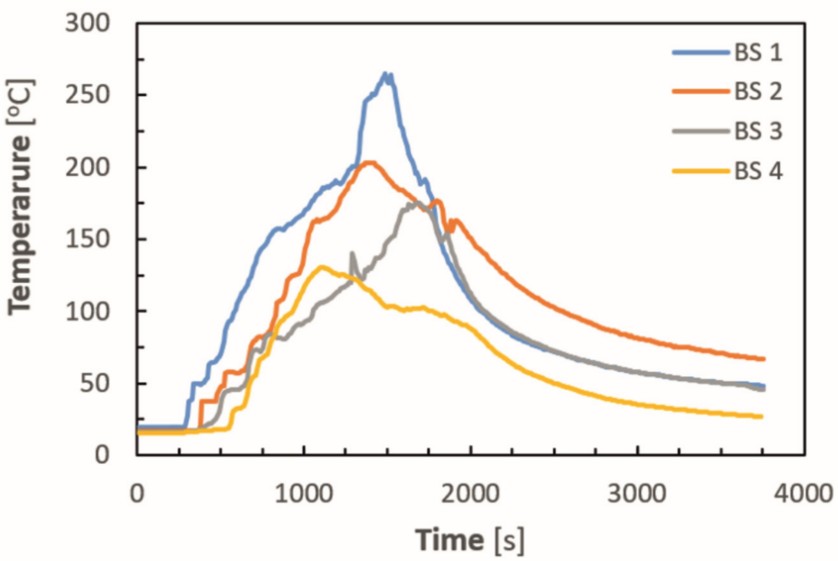

**Figure 6.** Average values of temperatures measured during the test in the insulation layer at level 2 (temperature measurement line L2, located 5 m above the combustion chamber).

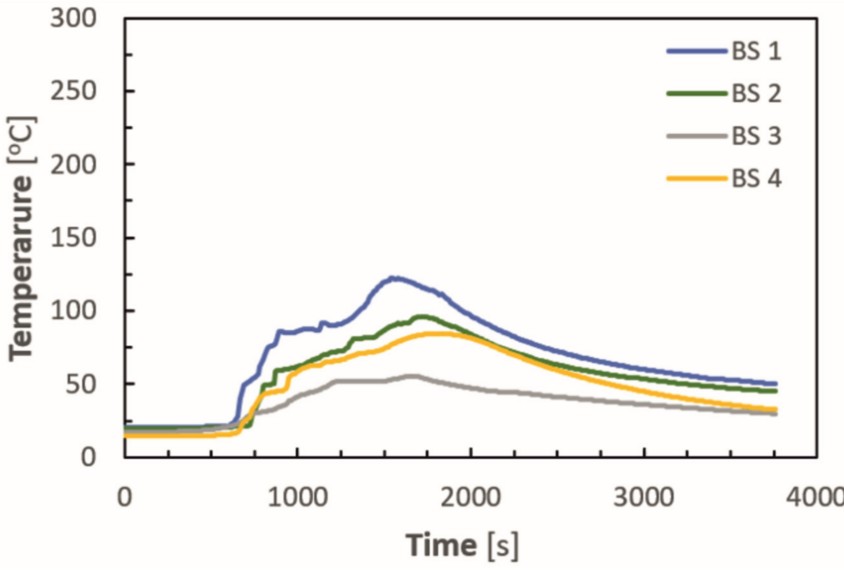

**Figure 7.** Average values of temperatures measured during the test in the insulation layer at level 3 (temperature measurement line L3, located 7.5 m above the combustion chamber).

Measurements of the average temperature inside the thermal insulation system at the level of 5 m and 7.5 m above the upper edge of the combustion chamber showed a faster temperature increase measured for variant I. At the same time, higher maximum temperatures, by 135 °C and 67 °C, occurred in the case of an ETICS-insulated wall compared to variant IV and III, respectively. The temperature difference corresponded with a slightly smaller range of polystyrene melting in variant III, in which MW was used in the lintel. For the wall with ETICS (variant I), polystyrene melting covered an area of approximately 34.8 m$^2$, and in the case of variant III, where this process was the smallest, approximately 31.4 m$^2$.

The analysis of the average temperature values obtained from all eight measurement points at the second line of thermocouples (L2) inside the insulation material shows that the maximum temperature value was obtained after 25 min of testing for variant I of a wall insulated with ETICS. For walls insulated according to variants II and IV, the maximum temperature was recorded earlier, after 24 and 19 min of the test, respectively. The slight decrease in the measured temperature that followed this time was caused by the burning of

polystyrene in the heat exposure from the fire source. The maximum average temperature value obtained for a wall insulated with variant III, at the same measurement level L2, shows a shift in time concerning a wall insulated with ETICS by 3 min. The differences in the values of the maximum temperatures at the individual levels are relatively small. The average temperature values achieved for both walls (insulated only with EPS and insulated with EPS with the use of MW lintel) at levels 2 and 3 confirm that no fire spread through the insulation layer. The temperature increase in both cases results from the fire source and the melting of polystyrene under the reinforced layer. The differences in the average temperatures measured inside the reinforced layer at the level of 5 m above the upper edge of the combustion chamber (measuring line L2) showed a comparable rate of temperature increase, and at the same time, the maximum value was higher by 52 °C in the case of variant II, which was obtained in the 21st minute of the test. For the level of 7.5 m, the analogous measurement showed a convergent rate of temperature increase for variants II and IV. However, due to the accumulation of heat below the second fire-protection zone, the maximum temperature was shifted towards variant IV, and the difference in maximum temperatures was 19 °C. The polystyrene melting surface covers a similar range (a difference of approximately 3 m$^2$), but it is worth noting that it has a different character in the case of the fourth variant.

Results showed that the maximum values of the temperatures obtained at individual points, on the L2 measuring line located at 5 m and the L3 measuring line at the height of 7.5 m above the upper edge of the combustion chamber, are higher by several dozen degrees than average values. The occurrence of these temporary temperatures over time is not reflected by temperature peaks in the case of the analysis of average temperature values. Although it clearly illustrates the system's behavior during fire development, the average temperature course cannot be used as a criterion for its assessment following BS 8414-1:2020 [10]. List of maximum temperature and the height of 5 and 7.5 m above the combustion chamber are presented in Table 2.

**Table 2.** List of maximum temperatures, along with the time of occurrence, recorded at individual measurement points located in the thermal insulation material at the height of 5 and 7.5 m above the combustion chamber.

| Measured Parameter | 2nd Level (Insulation Material) | 3rd Level (Insulation Material) |
|---|---|---|
| Ist wall [°C] | 364.5 | 177.2 |
| $T_{max}$ [s] | 1130.0 | 1540.0 |
| IInd wall [°C] | 260.2 | 156.2 |
| $T_{max}$ [s] | 960.0 | 1660.0 |
| IIIrd wall [°C] | 322.8 | 69.2 |
| $T_{max}$ [s] | 1170.0 | 1640.0 |
| IVth wall [°C] | 202.0 | 118.7 |
| $T_{max}$ [s] | 780.0 | 1590.0 |

The visual observations showed the necessity to carry out a thermal imaging analysis of the insulated walls after the fire source was extinguished. In variant II–IV, flame combustion of molten polystyrene on MW belts was recorded (Figure 8a–d). In the case of variant IV, a dangerous phenomenon was observed, which is the transition of MW into a state of continuous smoldering.

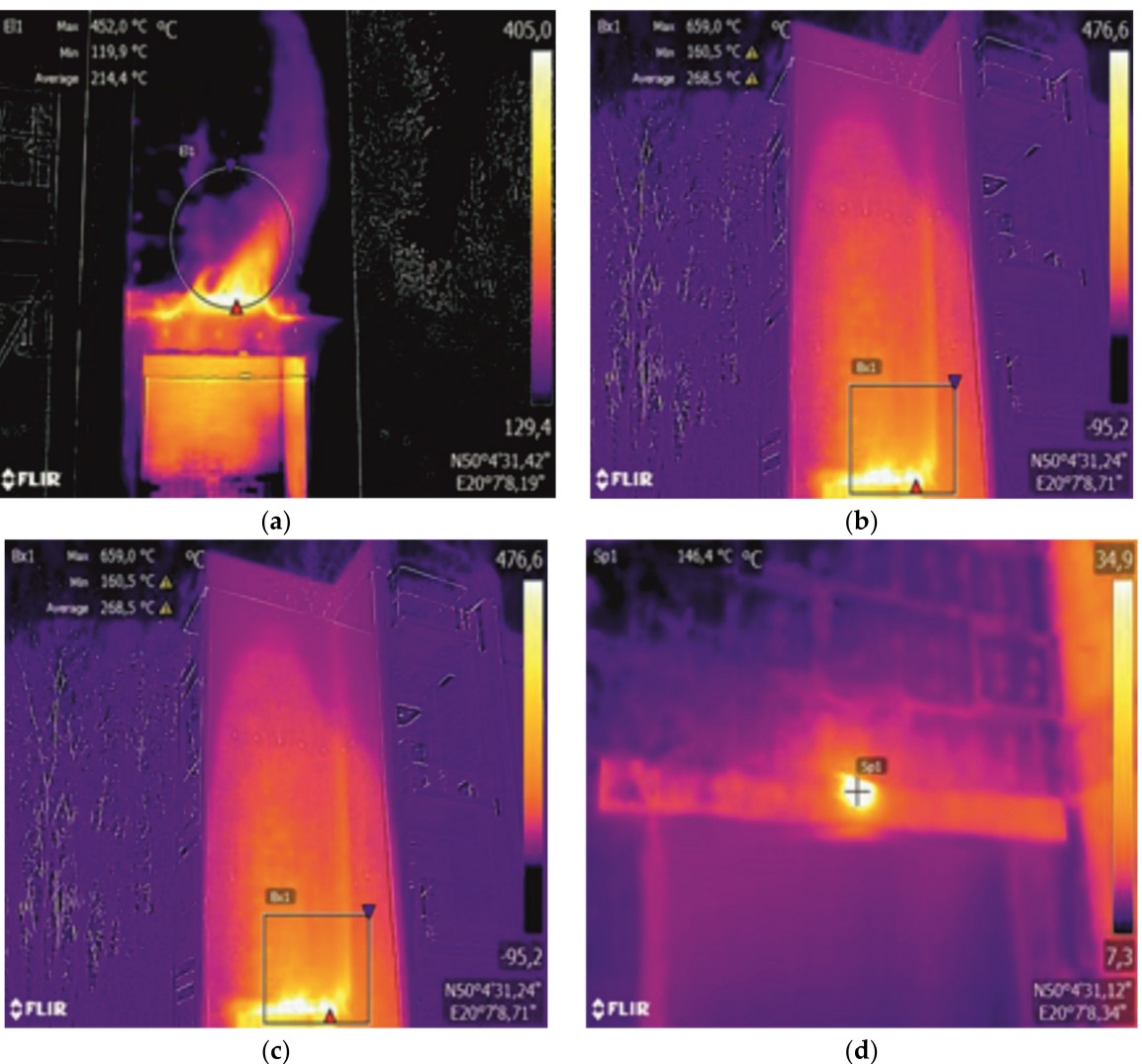

**Figure 8.** Thermal imaging photos of the tested thermal insulation systems after the end of the test, i.e., extinguishing the fire source: (**a**) burning melted polystyrene on a strip of MW located 40 cm above the edge of the fire source chamber—variant II; (**b**) combustion of melted polystyrene on the MW lintel—variant III; (**c**) burning molten polystyrene on the MW lintel—variant IV; (**d**) lintel in variant IV of thermal insulation 24 h after extinguishing the fire source—a transition of the material into a state of continuous smoldering.

## 4. Conclusions

### 4.1. Fire Performance of Investigated Facade Variants

For all tested variants, the criteria specified in the assessment and classification requirements of BR 135 [31], concerning the spread of fire through external walls, were fulfilled [8]. Following BS 8414-1: 2020 [10], the ETICS (variant I) was classified as a fire retardant, regardless of the MW partitions used. The study showed slight differences in average temperatures and the size of the polystyrene melting area for various insulation variants with the introduced horizontal MW partitions. The research did not fully confirm the significant benefits and the improvement of fire safety resulting from the application of MW partitions (concerning the variants described in the study). The fire safety and durability of the considered systems will be influenced, among others, by the correct execution of thermal insulation and strongly depended on the processing and protection of the combustion chamber lintel.

### 4.2. Fire Safety and Sustainable Use of Natural Resources

The aspects discussed in the introduction to this article, i.e., energy-saving production, low emission, and limited use of natural resources [4,5] strongly favor and support the use of sole EPS for the thermal insulation of buildings. From a holistic point of view, considering the significantly different environmental impact of ETICS with EPS and ETICS with MW, the obtained results should be considered [32]. The results, which do not show the advantage of using MW protective barriers on the facade of ETICS, and due to other basic requirements, only complicate the system, including the dimension of environmental impact. The reuse or recyclability of construction materials from the facade with EPS and MW is either impossible or very difficult [33]. In addition, considering the durability of the facade with EPS and MW, all connections between insulation materials make the system less durable [34].

### 4.3. Future Outlook of Experimental Results

It is necessary to continue research on this subject to verify and compare the results and obtain average values. One has to notice that the studies were not repeated, which may lead to uncertain results. Moreover, these tests are valid only for the selected geometries and the selected location of the fire barrier. To replicate experimental results, investigate selected geometries and locations of the fire barriers, and what is more to confront experimental results from different experimental methods, e.g., SP Fire 105 [14] and ISO 13785-2 [15], numerical models based on input data from experiments, enriched with heat release rate (HRR) measurements, should be used. Future research would allow the assessment of:

- The impact of fire barriers placed around openings (such as windows) and the difference between the behavior of ventilated facades for different variants of fire barriers (both vertical and horizontal);
- The possibility of hazards related to joining two insulation materials in one system, both due to the durability of the connection, i.e., the possibility of the discontinuity of the surface layer as a result of exploitation and the long-term impact of weather conditions;
- The occurrence of flame combustion of EPS in the place of application of vertical baffles made of MW, and the possibility of the MW turning into a state of continuous smoldering.

**Supplementary Materials:** The following supporting information can be downloaded at: http://systemyocieplen.pl/doc/sprawozdanie_z_badan_sciana_nr_1.pdf; http://systemyocieplen.pl/doc/sprawozdanie_z_badan_sciana_nr_2.pdf; http://systemyocieplen.pl/doc/sprawozdanie_z_badan_sciana_nr_3.pdf; http://systemyocieplen.pl/doc/sprawozdanie_z_badan_sciana_nr_4.pdf.

**Author Contributions:** Conceptualization, M.N., M.W. and K.B.; methodology, M.N., M.W. and K.B.; software, M.N., M.W. and K.B.; validation, M.N., M.W. and K.B.; formal analysis, M.N., M.W. and K.B.; investigation, M.N., M.W. and K.B.; resources, M.N., M.W. and K.B.; data curation, M.N., M.W. and K.B.; writing—original draft preparation, M.N., M.W. and K.B.; writing—review and editing, M.N., M.W. and K.B.; visualization, M.N., M.W. and K.B.; supervision, M.N., M.W. and K.B.; project administration, M.N., M.W. and K.B.; funding acquisition, M.N., M.W. and K.B. All authors have read and agreed to the published version of the manuscript.

**Funding:** This research was funded by the Polish Association for ETICS (SSO).

**Informed Consent Statement:** Not applicable.

**Conflicts of Interest:** The authors declare no conflict of interest. The funders had no role in the collection, analyses, or interpretation of data and in the writing of the manuscript.

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
