# Peer review of "Fire Safety of External Thermal Insulation Systems (ETICS) in the Aspect of Sustainable Use of Natural Resources"

_sustainability, doi:10.3390/su14031224_

Round 1
Reviewer 1 Report
The author's test method is relatively simple, possibly due to the limited research conditions. Nevertheless, the research content of this paper is very good and has strong practicability.
1. The sources of raw materials used in the experiment should be listed in the paper, especially the production company.
2. The conclusion part needs to be revised to clearly express the research results of this paper. Reduce your outlook for future work.
3. For actual combustion, testing is always very difficult. In the process of further research, it is suggested that the author should consider adding some content of computer simulation to analyze and interpret the actual test results.
Author Response
Dear Ms./Mr. Reviewer,
Thank you for the evaluation of the article entitled "Fire safety in the aspect of sustainable use of natural resources” and valuable comments provided. Below you will find answers to all of them.
- “The sources of raw materials used in the experiment should be listed in the paper, especially
the production company.”
Unfortunately, according to the agreement between the producer of raw materials (ETICS system components), The Polish Association of ETICS, and Research Network Łukasiewicz – Institute of Ceramics and Building Materials, we cannot publish raw materials trade names in the manuscript. Thus, we added the line in the manuscript that refers to the description of the investigated system and its components, European Technical Assessment, ETA no. 15/0582 issued in 2015 by Technical Assessment Body - Institute of Ceramics and Building Materials based on ETAG 004 used as EAD.
[new line 144] The investigated ETICS system (variant I) and its components (variant II-IV) were placed on the market according to European Technical Assessment ETA 15/0582 issued by the Institute of Ceramics and Building Materials based on ETAG 004 [25].
[new citation 25] Institute of Ceramics and Building Materials, European Technical Assessment (ETA) ETA 15/0582, External Thermal Insulation Composite Systems (ETICS) with rendering, Cracow, Poland, 2015,
- “The conclusion part needs to be revised to clearly express the research results of this paper. Reduce your outlook for future work.”
Thank you for the comment. It is difficult to reconcile two dissenting opinions to underline better future research plans (Reviewer #1) and reduce outlook for future works (Reviewer #2). Nevertheless, we decided to separate conclusions into the sections (5.1. Fire performance of investigated façade variants) and (5.2. Fire safety and sustainable use of natural resources) and (5.3. Future outlook of experimental results).
[new line 355] 5.1. Fire performance of investigated façade variants.
For all tested variants, the criteria specified in the assessment and classification requirements of BR 135 [31], concerning the spread of fire through external walls, were fulfilled [8]. Following BS 8414:2020-04, the ETICS (variant I), was classified as a fire retardant, regardless of the MW partitions used. The study showed slight differences in average temperatures and the size of the polystyrene melting area for various insulation variants with the introduced horizontal MW partitions. The research did not fully confirm the significant benefits and the improvement of fire safety resulting from the application of MW partitions (concerning the variants described in the study). The fire safety and durability of the considered systems will be influenced, among others, by the correct execution of thermal insulation and strongly depended on the processing and protection of the combustion chamber lintel.
5.2. Fire safety and sustainable use of natural resources
The aspects discussed in the introduction to this article, i.e., energy-saving production, low emission, and limited use of natural resources [4,5] strongly favor and support the use of sole EPS for thermal insulation of buildings support. And from the holistic point of view, considering the significantly different environmental impact of ETICS with EPS and ETICS with MW, the obtained results should be considered [32]. The results, which do not show the advantage of using MW protective barriers on the facade of ETICS with, and due to other basic requirements, only complicate the system, including the dimension of environmental impact. The reuse of recyclability of construction materials from the facade with EPS and MW is either impossible or very difficult [33]. Also, considering the durability of the facade with EPS and MW, all connections between insulation materials make the system less durable [34].
5.3. Future outlook of experimental results
It is necessary to continue research on this subject to verify and compare the results and obtain average values. One has to notice that the studies were not repeated, which may lead to uncertain results. Moreover, these tests are valid only for the selected geometries and the selected location of the fire barrier. To replicate experimental results, investigate selected geometries and locations of the fire barriers, and what is more to confront experimental results from different experimental methods, e.g. SP Fire 105 and ISO 135785-2, numerical models based on input data from experiments, enriched with heat release rate (HRR) measurement, should be used. Future research would allow the assessment of:
- the impact of fire barriers placed around openings (such as windows) and the difference between the behavior of ventilated facades for different variants of fire barriers (both vertical and horizontal).
- the possibility of hazards related to joining two insulation materials in one system, both due to the durability of the connection, i.e., the possibility of discontinuity of the surface layer as a result of exploitation and long-term impact of weather conditions;
- the occurrence of flame combustion of EPS in the place of application of vertical baffles made of MW and the possibility of the MW turning into a state of continuous smoldering.
[new citation 32] Michalak J. External Thermal Insulation Composite Systems (ETICS) from Industry and Academia Perspective. Sustainability 2021, 13(24), 13705.
[new citation 33] Superti, V.; Forman, T. V.; Houmani, C. Recycling Thermal Insulation Materials: A Case Study on More Circular Management of Expanded Polystyrene and Stonewool in Switzerland and Research Agenda. Resources 2021, 10(10),104.
[new citation 34] de Freitas, S. S.; de Freitas, V. P. Cracks on ETICS along thermal insulation joints: case study and a pathology catalogue. Structural Survey 2016, 34(1) 57 – 72.
- “For actual combustion, testing is always very difficult. In the process of further research,
it is suggested that the author should consider adding some content of computer simulation
to analyse and interpret the actual test results.”
We fully agree and share the Reviewer's opinion for further research possibilities. To sum up the last remark we would like shortly refer to it.
In our opinion, the large-scale tests are the best reproduction of real fire cases observed in the case of ETICS facades. The preparation process of the experiment is crucial for their success. We recognize the differences between different national testing methods and factors affecting the repeatability and reproducibility of fire performance test methods on a large scale. (1) There are factors affecting the ETICS system. It includes system mounting precision, the thickness of ETICS particular layers, and conditioning of the system. (2) There are also factors affecting measurement itself, e.g., the initial weight of each wood crib used for the BS 8414-1 test, which is a fire source is the main factor for the consistent nominal heat output of 4500 MJ at a peak rate of (3.0±0.5) MW and other factors like external temperature and wind speed.
In the case of measured variants, every stage was monitored both by the Polish Association of ETICS and authors of the manuscript, which includes the measurement of thicknesses of ETICS particular layers, preparation and conditioning of ETICS components and layers according to the manufacturer guidelines. For the best repeatability and reproducibility, we controlled the wood crib weight. The wind speed changed in the range ±0.2 m/s, and temperature in the range ±4°C (from measurement to measurement). Observed changes were considered as non-influencing the test itself.
Unfortunately, without monitoring of heat release rate (HRR) in the experiment, which is not part of the BS 8141-1 standard test, that should be used in numerical modelling as an input to the simulation, there is no chance to reproduce results shown in the manuscript with satisfying correlation factor to experiment [new citation 18]. Thank You for pointing out this issue. We will use simulations to analyze and interpret further research. We added addition lines addressing the issue of numerical simulation of fire exposed facades as an answer to Reviewer’s #2 comments.
Best regards,
Dr. Małgorzata Niziurska
Reviewer 2 Report
Dear authors,
thanks a lot for giving me the possibility to review the paper named : "Fire safety in the aspect of sustainable use of natural resources". I like the paper and I find very interesting. However I suggest these improvements:
a) about introduction/literature, I find a lack of literature. I suggest improving it. What about simulation? There are mmany papers you can use to improve it. I suggest only one 10.15866/iremos.v10i1.11133 (it's interesting the simulation approach). Please consider this if you like. You can find more other papers, but I don't suggest anymore because it's not my cup of tea
b) about conclusions, you have to underline better the future research and the weakness (you use too few words..)
Summarizing, there is a lack in the introduction/literature you can improve and in the conclusions.
Author Response
Dear Ms./Mr. Reviewer,
Thank you for the evaluation of the article entitled "Fire safety in the aspect of sustainable use of natural resources” and valuable comments provided. Below you will find answers to all of them.
- “about introduction/literature, I find a lack of literature. I suggest improving it. What about simulation? There are many papers you can use to improve it. I suggest only one 10.15866/iremos.v10i1.11133 (it's interesting the simulation approach). Please consider this if you like. You can find more other papers, but I don't suggest anymore because it's not my cup of tea”.
We fully agree and share the Reviewer's opinion that many papers involve numerical simulations and modelling of fire safety and fire spread in buildings, but not so many of them touch upon the issue of outdoor testing of ETICS facades on a large scale. Nevertheless, after available reports and publications analysis, published mainly by RISE Research Institute of Sweden, we decided to modify the introduction section and enrich it with additional literature describing the addressed issue.
[Line 96] Also, some numerical modeling methods are available to assess and validate differences between various experimental methods designed for fire performance testing of existing cladding systems, e.g., BS 8414-1 [10],
SP Fire 105 [14], and ISO 13785-2 [15]. Comparing the experimental measurement with simulation results always meets some uncertainties arising from natural variations in parameters and the effect of ambient conditions
[16-18]. In the case of the BS 8414-1 method, those differences were characterized as heat exposure (weight of wood crib), wind, and climate conditions since the test are performed outdoors. With the use of numerical methods, good agreement between experimental data and the numerical model is observed when measured in the experiment heat release rate (HRR) is used as an input in the simulations, excluding the area close to the burning chamber, where higher temperatures in the numerical model were observed [18]. Also, the same parameters are important for the repeatability and reproducibility of experimental results in large-scale testing according to BS 8414-1.
[new citation 14] SP FIRE 105, External Wall Assemblies and Façade Claddings, Reaction to Fire, Swedish National Testing and Research Institute Fire Technology, Borås, Sweden, 1994,
[new citation 15] International Organization for Standards (ISO), ISO 13785-2:2002 Reaction-to-fire test for façades – Part 2: Large-scale test. International Organization for Standards (ISO), Geneva, Switzerland, 2002,
[new citation 16] Karlsson, B., Carlsson, J., Numerical Simulation of Fire Exposed Facades - An initial investigation, Report 3123, Department of Fire Safety Engineering Lund University, Sweden, 2001.
[new citation 17] Andersson, J., Boström, L., Jansson, R., Milanović, B., Fire Dynamics in Façade Fire Tests: Measurement Modelling, and Repeatability, Application of Structural Fire Engineering, 15-16 October, Dubrovnik, Croatia,
[new citation 18] Andersson, J., Boström, L., McNamee, R. J., Fire Safety of Facades, SP Rapport 2017:37, RISE Research Institutes of Sweden, Borås, Sweden, 2017,
- “about conclusions, you have to underline better the future research and the weakness (you use too few words..)”
Thank you for the comment. It is difficult to reconcile two dissenting opinions to underline better future research plans (Reviewer #1) and reduce outlook for future works (Reviewer #2). Nevertheless, we decided to separate conclusions into the three sections (5.1. Fire performance of investigated façade variants) and (5.2. Fire safety and sustainable use of natural resources) and (5.3. Future outlook of experimental results).
[new line 355] 5.1. Fire performance of investigated façade variants.
For all tested variants, the criteria specified in the assessment and classification requirements of BR 135 [31], concerning the spread of fire through external walls, were fulfilled [8]. Following BS 8414:2020-04, the ETICS (variant I), was classified as a fire retardant, regardless of the MW partitions used. The study showed slight differences in average temperatures and the size of the polystyrene melting area for various insulation variants with the introduced horizontal MW partitions. The research did not fully confirm the significant benefits and the improvement of fire safety resulting from the application of MW partitions (concerning the variants described in the study).
The fire safety and durability of the considered systems will be influenced, among others, by the correct execution of thermal insulation and strongly depended on the processing and protection of the combustion chamber lintel.
5.2. Fire safety and sustainable use of natural resources
The aspects discussed in the introduction to this article, i.e., energy-saving production, low emission, and limited use of natural resources [4,5] strongly favor and support the use of sole EPS for thermal insulation of buildings support. And from the holistic point of view, considering the significantly different environmental impact of ETICS with EPS and ETICS with MW, the obtained results should be considered [32]. The results, which do not show the advantage of using MW protective barriers on the facade of ETICS with, and due to other basic requirements, only complicate the system, including the dimension of environmental impact. The reuse of recyclability of construction materials from the facade with EPS and MW is either impossible or very difficult [33]. Also, considering the durability of the facade with EPS and MW, all connections between insulation materials make the system less durable [34].
5.3. Future outlook of experimental results
It is necessary to continue research on this subject to verify and compare the results and obtain average values. One has to notice that the studies were not repeated, which may lead to uncertain results. Moreover, these tests are valid only for the selected geometries and the selected location of the fire barrier. To replicate experimental results, investigate selected geometries and locations of the fire barriers, and what is more to confront experimental results from different experimental methods, e.g. SP Fire 105 and ISO 135785-2, numerical models based on input data from experiments, enriched with heat release rate (HRR) measurement, should be used. Future research would allow the assessment of:
- the impact of fire barriers placed around openings (such as windows) and the difference between the behavior of ventilated facades for different variants of fire barriers (both vertical and horizontal).
- the possibility of hazards related to joining two insulation materials in one system, both due to the durability of the connection, i.e., the possibility of discontinuity of the surface layer as a result of exploitation and long-term impact of weather conditions;
- the occurrence of flame combustion of EPS in the place of application of vertical baffles made of MW and the possibility of the MW turning into a state of continuous smoldering.
[new citation 32] Michalak J. External Thermal Insulation Composite Systems (ETICS) from Industry and Academia Perspective. Sustainability 2021, 13(24), 13705.
[new citation 33] Superti, V.; Forman, T. V.; Houmani, C. Recycling Thermal Insulation Materials: A Case Study on More Circular Management of Expanded Polystyrene and Stonewool in Switzerland and Research Agenda. Resources 2021, 10(10),104.
[new citation 34] de Freitas, S. S.; de Freitas, V. P. Cracks on ETICS along thermal insulation joints: case study and a pathology catalogue. Structural Survey 2016, 34(1) 57 – 72.
Best regards,
Dr. Małgorzata Niziurska